# Acute Spontaneous Lobar Cerebral Hemorrhages Present a Different Clinical Profile and a More Severe Early Prognosis than Deep Subcortical Intracerebral Hemorrhages—A Hospital-Based Stroke Registry Study

**DOI:** 10.3390/biomedicines11010223

**Published:** 2023-01-16

**Authors:** Joana Maria Flaquer-Pérez de Mendiola, Adrià Arboix, Luís García-Eroles, Maria José Sánchez-López

**Affiliations:** 1Department of Neurology, Hospital Universitari Sagrat Cor, Universitat de Barcelona, 08029 Barcelona, Spain; 2Medical Library, Hospital Universitari Sagrat Cor, Quirónsalud, 08029 Barcelona, Spain

**Keywords:** acute stroke, cerebrovascular diseases, dementia, intracerebral hemorrhage, in-hospital mortality, lobar hemorrhage, outcome, stroke, stroke registry, subcortical hemorrhage

## Abstract

Acute spontaneous intracerebral hemorrhage (ICH) is the most severe stroke subtype, with a high risk of death, dependence, and dementia. Knowledge about the clinical profile and early outcomes of ICH patients with lobar versus deep subcortical brain topography remains limited. In this study, we investigated the effects of ICH topography on demographics, cerebrovascular risk factors, clinical characteristics, and early outcomes in a sample of 298 consecutive acute ICH patients (165 with lobar and 133 with subcortical hemorrhagic stroke) available in a single-center-based stroke registry over 24 years. The multiple logistic regression analysis shows that variables independently associated with lobar ICH were early seizures (OR 6.81, CI 95% 1.27–5.15), chronic liver disease (OR 4.55, 95% CI 1.03–20.15), hemianopia (OR 2.55, 95% CI 1.26–5.15), headaches (OR 1.90, 95% CI 1.90, 95% IC 1.06–3.41), alcohol abuse (>80 gr/day) (OR 0–10, 95% CI 0.02–0,53), hypertension (OR 0,41, 95% CI 0.23–0–70), sensory deficit (OR 0.43, 95% CI 0.25–0.75), and limb weakness (OR: 0.47, 95% CI 0.24–0.93). The in-hospital mortality was 26.7% for lobar and 16.5% for subcortical ICH. The study confirmed that the clinical spectrum, prognosis, and early mortality of patients with ICH depend on the site of bleeding, with a more severe early prognosis in lobar intracerebral hemorrhage.

## 1. Introduction

Stroke remains a leading cause of death and disability and has a complex pathophysiology. According to the 2022 Heart Disease and Stroke Statistics Update published by the American Heart Association, there were 7.08 million deaths attributable to cerebrovascular disease worldwide in 2020, and stroke accounted for approximately 1 in 19 deaths in the United States in 2019, making it the fifth leading cause of death in the country [1,2]. Stroke is the third leading cause of death in most Western countries, after coronary heart disease and cancer [3,4,5].

According to different series, 10–15% of all acute strokes correspond to intracerebral hemorrhages (ICH). The approximate incidence of ICH is 24.6 cases/100,000 persons, and this figure increases with age, masculine sex, and among the Asian population [6,7,8,9,10,11]. Acute spontaneous ICH is the most devastating and severe stroke subtype, with a high risk of death, dependence, epilepsy, and dementia, posing a serious public health problem [12,13,14,15].

ICHs are bleedings located in the brain parenchyma and originate from a rupture of blood vessels, mostly of the arterial type [16,17]. ICH destroys the brain structure through the direct pressure effects of an acutely expanding mass lesion. Hematoma may increase intracranial pressure, causing oppression of brain areas that can potentially affect blood flow, resulting in brain edema and primary brain injury, and causing increased intracranial pressure, hydrocephalus, or herniation [17,18]. Secondary physiological and cellular injuries from it include inflammation and the biochemical toxicity of blood products, such as hemoglobin, iron, and thrombin [19,20,21].

There is a paucity of clinical studies focusing specifically on the subtypes of ICH according to brain location, which is a topic of interest to clinicians involved in stroke management.

Spontaneous or non-traumatic ICH is usually classified as primary ICH when caused by the rupture of small brain vessels damaged by different forms of small vessel disease; or as secondary ICH when the hemorrhage is caused by structural cerebral vascular malformations, brain tumor, inherent coagulopathy, or other rare diseases [22,23,24].

It is widely accepted that small vessel diseases are the main causes, accounting for about 80% of spontaneous ICH [25,26]. This large subgroup mainly includes two sporadic entities: hypertensive angiopathy or deep perforating arteriolosclerosis, with a usual subcortical cerebral topography, and sporadic cerebral amyloid angiopathy (CAA), with a usual lobar cerebral localization. Other rare etiologic forms (familial CAA, CADASIL or COL4A1 mutations, or post-radiation angiopathy) are rare [25,26,27,28,29].

Secondary causes account for about 20% of spontaneous ICH due to arteriovenous malformation; dural arteriovenous fistula or cerebral malformations have been reported in 15.23% of adults with ICH and brain tumors in less than 5%. Other causes of ICH are summarized in Table 1 [30,31,32].

About 80% of patients with ICH present with elevated blood pressure on admission, and most of them have a history of hypertension [33,34,35]. However, there has been an increase in ICH among the young population caused mainly by drug abuse (cocaine, heroin, amphetamine, and ephedrine). Iatrogenic hemorrhages (caused by anticoagulant or thrombolytic drugs) should also be considered as they are another major etiology of hemorrhages nowadays [36]. All these etiologies fluctuate depending on many factors, mainly age. In young patients, vascular malformations and sympathetic drugs are the most common causes; among the middle-aged population, arterial hypertension predominates, and in patients over 70 years of age, amyloid angiopathy and iatrogenic bleeding are the most frequent [37].

The traditional view considers that the difference between hypertensive vascular disease and amyloid vascular disease lies in the site of bleeding. ICH associated with amyloid angiopathy is usually located in the lobe, and hypertensive ICH is usually located deep in the brain. However, hypertension can also lead to lobar hemorrhage, and hypertensive vascular disease and amyloid vascular disease may be a co-disease, so the pathogenesis of ICH is sometimes difficult to determine [35,36,37].

The onset of ICH is usually sudden (within minutes) or acute (within hours). In many cases, headaches, nausea, and a decreased level of consciousness are common [3,6]. The latter symptom, if persistent and leading to a state of poor reactivity, should alert clinicians to major bleeding in the cerebral parenchyma. Although ICH can produce seizures (focal or generalized) because of parenchymal irritation due to the presence of blood, these are rarely recurrent or lead to an epileptic status [38,39].

However, knowledge about the clinical profile and early outcomes of patients with ICH of lobar versus deep subcortical brain location remains limited. Depending on the topographic involvement, the clinical presentation is variable. Thus, the most common presentation is a contralateral paresis and hypoesthesia of the affected region, although depending on the affected area, symptoms may vary considerably. Previous studies have repeatedly noted that lobar hemorrhages debut mainly with headaches and seizures, whereas subcortical hemorrhages debut more frequently with sensory–motor deficits [30,32,40]. However, knowledge of the clinical profile according to the hemorrhagic brain topography remains limited, and the key factors affecting the different clinical profiles in intracerebral hemorrhage should continue to be studied in future research to aid in the development of effective interventional treatment measures.

Intracerebral hemorrhage accounts for approximately 34–52% of the 3-month mortality among patients with stroke, and only 20% of patients with ICH achieve functional independence within 6 months [30,31,32]. These rates are related to the space occupied by the hemorrhage in the cerebral parenchyma (the larger they are, the greater the mortality they entail, being maximal in hematomas larger than 60 cm^3^). The relationship between brain location and in-hospital mortality is controversial. The location of the lesions in the deep cerebral area present a higher mortality rate in some studies [13,32], but in others, the worst prognosis is for hemorrhages involving the brainstem or affecting multiple locations (multifocal). These hemorrhagic stroke topographies have a mortality of over 40% [41]. In contrast, capsular hemorrhages seem to have a better prognosis and less neurological involvement [42], while lobar topography presents inconclusive results.

Furthermore, when analyzing ICH globally, altered consciousness at stroke onset correlates well with ICH severity and prognosis, and limb weakness is a good clinical predictor of neurological sequelae [38,39]. There are also demographic factors or neuroimaging data, such as intraventricular bleeding, that worsen the prognosis of patients of an advanced age [30,32]. In lobar hemorrhages, the anticoagulation levels and delays in neuroimaging procedures also correlate with higher mortality rates [21]. However, the evidence to determine the relationship between in-hospital mortality or early outcomes and the different hemorrhagic topographies of stroke is still limited.

### Hypothesis and Objectives

ICH encompasses a variety of hematomas in different locations and sizes. The main objective of this retrospective, single-center, comparative clinical study was to extend and update the knowledge on the poorly understood relationship in the two main topographic subtypes of spontaneous ICH (lobar versus deep subcortical) on clinical profiles, early outcomes, and prognosis, using data from a single-center stroke registry ongoing for 24 years.

## 2. Materials and Methods

### 2.1. Study Population

We present a study, conducted at the Sagrat Cor University Hospital of Barcelona, based on the analysis of the stroke database registered for 24 years (from 1985 to 2009, both included). This registry has been previously published and validated [39].

Stroke subtypes, cardiovascular risk factors, and clinical and etiological features were classified according to the guidelines of the Cerebrovascular Disease Study Group of the Catalan Society of Neurology, Cerebrovascular Study Group of the Spanish Society of Neurology, which is similar to the National Institute of Neurological Disorders and Stroke Classification and has been used by our group in previous studies [43,44,45].

According to these guidelines, spontaneous intracerebral hemorrhages are defined as the bleeding caused by a nontraumatic rupture of the blood vessels (mostly arterial type) directly to the brain parenchyma and are considered different entities from subarachnoid hemorrhages, subdural or epidural hematomas, or secondary hemorrhages in the pons (Duret’s hemorrhages) caused by sudden pressure changes that distort and compress the brainstem, which had previously been excluded.

The location of the intracerebral hemorrhage was identified on non-contrast computed topography (CT) scans of the brain and/or magnetic resonance imaging. Neuroimaging data were necessary to rule out stroke mimics, confirm the clinical diagnosis, and distinguish ischemia from ICH. MR angiography, CT angiography, and digital subtraction angiography may be considered if clinical suspicion of vascular malformations is high and noninvasive studies do not show a clear cause, especially in young normotensive and surgical candidates.

From the initial sample of 408 patients diagnosed with spontaneous intracerebral hemorrhages, those with topographies in the cerebellum (28), brainstem (18), multiple topographies (51), primary intraventricular hemorrhages (12,) and spinal cord hemorrhages (1) were excluded.

Thus, the hemorrhages were classified into lobar topography type when affecting cerebral lobes (Figure 1A) and deep subcortical topography type (Figure 1B), when the latter composed of the lateral basal-ganglionic-capsular region and/or thalamic cerebral location.

Thus, a total sample of 298 patients (n = 298) was obtained, of which 165 belonged to the lobar hemorrhage group, while the remaining 133 corresponded to subcortical hemorrhages (50 thalamic and 83 capsuloganglionic), as can be seen in Figure 2.

All patients were admitted to hospital within 48 h of the onset of symptoms. The following information was recorded for each patient: demographic characteristics, vascular risk factors, clinical variables, results of routine laboratory tests, chest radiography, 12-lead electrocardiography, neuroimaging data (non-contrast CT, CT angiography, or MRI), and outcomes. Other investigations included arterial digital subtraction angiography in 25% of patients, echocardiography in 10%, lumbar puncture in 6%, and immunologic testing in 3%. The definitions of vascular risk factors were those used in previous studies [46,47]. A patient was considered to have chronic liver disease if he/she had been previously diagnosed by the Gastroenterology Service of our center with liver cirrhosis or chronic hepatitis. In five patients presenting with lobar clots >30 mL and with 1 cm of surface area, surgical evacuation of supratentorial ICH by standard craniotomy was performed. Causes of death were analyzed according to the criteria of Silver et al. [48,49]. The frequency of patients with complete (no clinical disability) or good (score 0–2 in the modified Rankin scale) outcomes at hospital discharge was also assessed [50]. The aim of this clinical study was to assess the differential characteristics in etiology, risk factors, clinical features, and early prognosis between cohorts of patients with lobar hemorrhages and subcortical hemorrhages. Prior to conducting the study, approval was obtained from the Ethical Committee of Clinical Research of the hospital.

### 2.2. Statistical Analysis

A comparative analysis was performed between patients with lobar hemorrhages (n = 165) versus subcortical hemorrhages (n = 133). Univariate analysis for each variable in relation to demographics, cerebrovascular risk factors, clinical manifestations, and outcomes was assessed using the chi-square test, the Student’s *t*-test, and the Fisher’s exact test, and statistical significance was set at *p* < 0.05. Variables were subjected to multivariate analysis with a logistic regression procedure and forward stepwise selection if *p* < 0.2.

The independent predictive value of each variable on lobar topography (versus subcortical location) was studied in three multiple regression models. The first one was based on demographic and risk factor data, with 5 variables (female sex, hypertension, previous intracerebral hemorrhage, alcohol abuse, and chronic liver disease). The second model was based on demographical data, risk factors, and clinical features, with 10 variables (hypertension, previous intracerebral hemorrhage, alcohol abuse, chronic liver disease, headaches, early seizures, altered consciousness, limb weakness, sensory disorders, and hemianopia). The last model was based on demographics, risk factors, clinical features, and early outcomes, with 8 variables (hypertension, alcohol abuse, chronic liver disease, headaches, early seizures, limb weakness, sensory disorders, and hemianopia).

In the multivariate analysis, the cerebral topography (coded as subcortical location = 0 and lobar location = 1) was the dependent variable. The level of significance required to remain in the models was 0.15. The tolerance level was established as 0.0001. The maximum likelihood approach was used to estimate the weights of the logistic parameters [51]. Odds ratio and 95% confidence intervals (CI) were calculated from the β-coefficients and standard errors. The hypothesis that the logistic model adequately fits the data was tested using the goodness-of-fit chi-squared test. The area under the receiver operating characteristics (ROC) curve was calculated for each predictive model. The SPSS-PC + and BMDP software were used for statistical analysis [52].

## 3. Results

### 3.1. General Data

The study population included 298 consecutive patients with acute spontaneous ICH, diagnosed with lobar or subcortical ICH. Lobar cerebral hemorrhages were present in 165 patients (55.7%) and classic deep subcortical intracerebral hemorrhages (thalamic or capsuloganglionar) in 133 patients (44.3%) (Figure 3).

The percentage of women was higher in lobar hemorrhages (53.9%), while males were more frequent in subcortical hemorrhages (61.7%). The demographic characteristics of the patients included in the study are shown in Table 2.

The cause of ICH was hypertension in 46% of cases, arteriovenous malformation in 7.5%, anticoagulation in 3%, and hematological alterations in 2.5%. The cause of bleeding was not identified in 41%.

Sixty-six patients died (in-hospital mortality 22.1%). The causes of death included cerebral herniation in forty patients, pneumonia in eight, sepsis in eight, sudden death in three, myocardial infarction in one, pulmonary thromboembolism in one, and unknown causes in five.

### 3.2. Differences between Lobar Cerebral Hemorrhages and Deep Subcortical Intracerebral Hemorrhages

In the comparative analysis between the two subtypes (Table 3), it was observed that AHT, obesity, and alcoholism were significantly higher within the subcortical group. Moreover, previous cerebrovascular disease, anterior cerebral hemorrhages, a history of headaches, and the presence of liver disease were significantly more frequent among lobar hemorrhages.

Regarding the clinical features of the current episode, in the subcortical group, there was a higher rate of instantaneous debut (in minutes), while in the lobar subtype, the most frequent onset was acute (in hours). The clinical manifestations that significantly predominated among lobar hemorrhages were headaches, dizziness, seizures, consciousness disorders, visual disturbances, and praxis disorders. As for the symptoms of the subcortical hemorrhages, the more frequent were motor disorders, sensory disorders, and cranial pair alterations.

Regarding early outcomes, the frequency of in-hospital mortality was significantly higher in lobar hemorrhages (26.7%) compared to subcortical hemorrhages (16.5%). The absence of symptoms at hospital discharge was unusual in both the locations of ICH (6.1% vs. 8.3%, respectively).

### 3.3. Multivariate Analysis

In the multivariate analysis (Table 4), chronic liver disease was the only variable that was independently associated with lobar hemorrhage in the three logistic regression models. The variables selected in the third model based on the demographic data, cerebrovascular risk factors, clinical characteristics, and in-hospital evolution were chronic liver disease (OR 4.551; IC 1.027–20.154; *p* = 0.046), headaches (OR 1.904; IC 1.062–3.414; *p* = 0.031), early seizures (OR 6.806; IC 1.273–36.397; *p* = 0.025), and hemianopia (OR 2.551; IC 1.264–5.149; *p* = 0.009). Variables that proved to be independent predictors of deep intracerebral subcortical hemorrhages were hypertension (OR 0.406; IC 0.234–0.704; *p* = 0.001), alcohol abuse (OR 0.098; IC 0.018–0.528; *p* = 0.007), limb weakness (OR 0.475; IC 0.243–0.928; *p* = 0.029), and sensory disorders (OR 0.434; IC 0.248–0.761; *p* = 0.004).

Figure 4 shows the ROC curve for the accuracy of the regression model, including demographic data, cerebrovascular risk factors, clinical features, and outcomes. The area under the curve was 0.785. The sensitivity was 66.7%, the specificity was 72.9%, the positive predictive value was 75.3%, and the negative predictive value was 63.8%.

## 4. Discussion

Spontaneous ICH is a devastating stroke subtype with high rates of mortality and long-term disability. The main causes of non-traumatic ICH in adults are hypertension, cerebral amyloid angiopathy, and anticoagulation [30,31,32]. The influence of the bleeding site on the clinical spectrum and outcomes of ICH patients remains a poorly defined aspect of the disease, to which should be added that lobar ICH is difficult to differentiate from subcortical ICH at the bedside.

However, the comparative analysis performed in our study suggests markedly different data on the demographics, risk factors, clinical features, and early outcomes. Finally, brain imaging is essential to distinguish the location and determine the ICH volume [53].

In our experience, after analyzing the data collected in our acute stroke registry for 24 consecutive years, lobar ICH, in which the hematoma is located in the frontal, parietal, temporal, or occipital lobe matter, was more frequent (55.7%) than deep ICH (44.3%), which is similar to what has been observed in other recent studies [11,17,18,19].

There is a paucity of information on the demographic characteristics between patients with lobar versus subcortical ICH. This investigation evidences the existence of differences between lobar and subcortical hemorrhages. Regarding age, a mean of 72 years was present in both subtypes of the ICH location. However, the presence of very old patients (85 years or older) in 16.4 and 14.3% of lobar and subcortical ICHs stands out. This fact had already been observed in studies analyzing acute stroke in general, but this information on ICH is limited [54,55,56]. Due to the fact that life expectancy is expected to increase, the mean age of this disease is also expected to rise in the coming years. Consequently, patients affected by ICH will have more comorbidities, and their management could become even more complex. This fact will entail a challenge for future physicians due to the higher frequency of ICH in the oldest old patients.

We found that hypertension, the most important cerebrovascular risk factor, in agreement with other studies, was an independent predictive risk factor for deep subcortical hemorrhages. Hypertensive cerebral vasculopathy is more frequent in deep brain structures as a consequence of damage to the walls of the small cerebral blood vessels traversing these regions due to hypertension causing deep subcortical ICH [57].

Alcohol abuse is another risk factor associated with deep subcortical ICH. Alcohol abuse injures the endothelial integrity of the deep perforating arteries, which is the main cause of ICH located in the basal ganglia and thalamus. Chronic alcoholism also increases the presence of hypertension. Furthermore, alcohol potentiates the arteriosclerotic process, especially among the small deep subcortical vessels of the brain [58].

However, lobar ICH was less frequently associated with hypertension. Thus, the non-hypertensive mechanisms of ICH predominate in the lobar location; these are mainly cerebral amyloid angiopathy, vascular malformations, sympathomimetic drugs, and bleeding disorders. Cerebral amyloid angiopathy is the most frequently associated with lobar ICH and is the result of β-amyloid deposition in arterioles and capillaries, meaning the leptomeninges and cortical vessels [7,8,9].

The main independent risk factor associated with lobar hemorrhages was chronic liver disease. There is little information on the relationship between liver function and ICH with lobar hematoma. The pathophysiological mechanism explaining the association between liver disease and ICH is coagulopathy and thrombocytopenia secondary to liver damage leading to a bleeding tendency in these patients [58,59]. Our results agree with the study by Hoya et al., in which the ratio of non-deep cerebral hemorrhages was significantly higher in the patients with chronic liver disease [58]. Cooperation with hepatologists is important when dealing with ICG accompanied by chronic liver disease, as these patients are at greater risk of non-neurological complications than idiopathic ICH patients.

The risk of ICH recurrence is increased in lobar hemorrhage secondary to amyloid angiopathy. In this setting, the recurrent intracerebral hemorrhagic episodes are a predictor or vascular cognitive impairment or vascular dementia [30].

There is scant information on the differences in the clinical spectrum and outcomes of hemorrhagic stroke depending on the site of hemorrhage. Patients with lobar ICH reveal the significantly more frequent presence of headaches, early seizures, and hemianopia.

In the previous clinical studies of our research group, we have shown that early seizures are more frequent in hemorrhagic than acute ischemic stroke, and, in ICH patients, early epileptic seizures are more frequent in lobar than deep cerebral hematomas, mainly in the parietal and temporal lobes, which are the most epileptogenic locations [32,60]. However, the relationship between seizures, functional outcomes, and mortality is complex and not well-defined.

Hemianopia is related to lobar hemorrhages of occipital topography or by a lesion of the optic radiations running through the temporal and parietal lobes. As for the headache, it is due to the fact that hematoma causes local distention and inflammation of the parenchyma where it is located. It is also known that, in occipital hemorrhages, headaches can be explained by the organization of the trigeminovascular system [61].

In addition, sensitive and motor disorders appear to be independent predictors of the subcortical subtype. Prominent sensory loss, either anesthesia or hypoesthesia affecting the face, limbs, and trunk, was found in 70% of patients, generally for all sensory modalities, which is consistent with classical reports showing the predominance of sensory deficit as a cardinal feature of thalamic hemorrhage, especially when the ventroposterolateral nucleus is affected [16,17,18]. Other typical symptoms of thalamic hemorrhages are oculomotor signs and language disturbances (mostly when the lesion is located in the dominant hemisphere) [27]. When the posterior branch of the internal capsule is involved, predominantly the motor or mixed motor and sensitive symptoms are more frequent, and the clinical presentation of deep subcortical hemorrhages in the form of a lacunar syndrome (pure sensory stroke, sensorimotor stroke, or pure motor hemiparesis) is documented in 6–8% of cases [62].

As for alterations in the level of consciousness, they were shown to be associated with lobar hemorrhages in the univariate analysis, but this correlation was not statistically significant in the multivariate analysis. This is probably because a decrease in the consciousness level has always been found to be a predictor of mortality in the different series and appears frequently associated with increased intracranial pressure and compression of the diencephalon and brainstem or occurs if there is a multiple topography bleeding [14,15]. Multiple topography hemorrhages were not included in this study, and this may explain why no significant correlation with this variable was appreciated. Consciousness level disorders also appear when the brainstem is affected, and this is due to the fact that the brainstem contains the ascending reticular activating system. As, in this work, hemorrhages of this topography were also excluded, no correlation has been observed.

The absence of neurological deficit at discharge is uncommon in both lobar (6.1%) and subcortical hemorrhage patients (8.3%). However, lobar hemorrhages have a higher in-hospital mortality (26.7%) than deep subcortical ones (16.5%), as observed in the preliminary studies by our research group [19]. This is observed despite the fact that infectious (25.6% vs. 15.2%) and renal (4.5% vs. 0.6%) complications were more frequent in subcortical than lobar hemorrhages.

One hypothesis is that early epileptiform activity, more frequent in lobar hemorrhages, has a deleterious effect on injured brain areas, perhaps due to an anoxic event, and contributes to secondary brain damage [63,64]. This high susceptibility of cerebral lobar involvement to seizures is consistent with the finding of other authors [65,66]. There is, however, little experimental evidence between post-stroke epileptiform activity and neuronal loss. Tan et al. [67] demonstrated that the suppression of epileptiform activity with MK-80 reduced neuronal loss and slowed the development of secondary edema after a global hypoxic-ischemic insult in rats. On the other hand, post-stroke seizures may be detrimental to the brain by increasing energy metabolism. Cerebral blood flow and glucose and oxygen consumption increase substantially during generalized seizures to meet the increased energy needs of the tissue. Likewise, another neuropathological study reported hemispheric brain damage secondary to unilateral status epilepticus [68,69].

Chronic liver disease, also more frequent in lobar hemorrhage, is associated with in-hospital mortality and an unfavorable discharge disposition after ICH. Liver fibrosis is independently associated with 90-day mortality in the study of Parikh et al. [59]. This could reflect a propensity for increased bleeding due to the hemostatic disorders in chronic liver diseases, as evidenced by higher ICH volumes on admission and hematoma expansion. Increased susceptibility to infections and liver-related complications are alternative reasons for higher mortality after ICH [59,60].

Another reason for higher mortality in lobar hemorrhages could be due to the fact that the hematoma volume is usually smaller in deep subcortical hemorrhages than lobar hemorrhages. The volume of the clot at the time of hospital admission is a good predictor of early functional outcomes and, therefore, the hemorrhage volume was another predictor of mortality for all locations in acute spontaneous ICH [1,2,14,70].

Most of our ICH patients were treated medically, so this study contributes to the knowledge of the natural history of patients with acute spontaneous ICH of the lobar versus deep subcortical location in referral centers. The optimal treatment strategy for ICH is effective prevention and long-term monitoring for high-risk groups, as well as internal medicine support. Unfortunately, there is still no optimal surgical treatment for acute ICH that is clearly beneficial, and the role of the removal of hematoma with minimal trauma surgery remains controversial despite its widespread use in various forms [32].

In a clinical study, Sturiale et al. [71] highlighted the possibility that in-patients with hemorrhagic brain arteriovenous malformation and non-parenchymal bleeding (subarachnoid hemorrhage and intraventricular hemorrhage) may worsen the prognosis of patients with hemorrhagic arteriovenous malformations.

In summary, as long as ICH is a disease lacking a specific and effective treatment, primary prevention could be a powerful tool for its control. The identification of risk factors that contribute to the genesis of these brain hemorrhages could allow us to reduce their incidence and, therefore, decrease the resulting morbidity and mortality rates.

This study has some limitations. Firstly, it is a retrospective, cross-sectional clinical study conducted at a single center. In this regard, we recommend that future large multicenter studies be carried out. Likewise, further validation of our results in a new prospective clinical study would be of interest. A second limitation of the study is that we have not analyzed the prognostic value of ICH in the medium or long term.

Another limitation of the study is that secondary bleeding due to vascular malformations (7.5% of the sample) was included in the early outcomes. However, treatment and early prognosis in this group of patients are very different, as bleeding due to rupture of a cavernous malformation is less life-threatening (and treatment is usually surgical with better outcomes) than macrovascular lesions such as arteriovenous malformations and aneurysms [30].

In future studies, these aspects would be interesting lines of research. However, the assessment of the methods used in this study based on the results of a stroke database of a large sample of 504 acute hemorrhagic stroke consecutive patients collected over a period of 24 years is more than less objective. Furthermore, a future line of investigation would be the relevance and evidence of cerebral atrophy associated with disability and vascular cognitive impairment in ICH patients [12,72].

## 5. Conclusions

Acute spontaneous lobar and deep subcortical ICH are severe diseases. The cerebrovascular risk factors for each topographic subtype of ICH are different. Thus, hypertension and alcohol abuse have been found to be more related to deep subcortical hemorrhages, whereas chronic liver disease predisposes to lobar hemorrhages. This would allow the realization of a major primary prevention of these diseases by reducing the incidence of these cerebrovascular risk factors by treating them appropriately.

Although the definitive distinction between lobar and subcortical hemorrhages is made by neuroimaging tests, in turn, early seizures, headaches, and hemianopia are associated with lobar hemorrhages, whereas motor and sensory disorders are more frequent among subcortical hemorrhages.

Morbidity and mortality associated with ICH remain high despite recent advances in our understanding of the clinical course of ICH: in-hospital mortality was 26.7% for lobar and 16.5% for subcortical ICH, with the occasional absence of neurological deficit at hospital discharge observed (6.1% vs. 8.3%, respectively). The association between early seizures and chronic liver disease, with acute lobar hemorrhages, could explain this unfavorable early outcome.

In summary, the lobar and deep subcortical topographies of ICH influence the clinical spectrum and early outcomes. Rapid recognition and diagnosis of the ICH location, as well as identification of the early prognostic indicators, are essential to plan the level of care and avoid acute rapid progression during the first hours. This could be a very useful tool to reduce the sequelae and mortality rates caused by these catastrophic life-threatening diseases.

ICH is a complex clinical event that has been shown to benefit from specially trained multidisciplinary care. Trials of the minimally invasive surgical evacuation of clots according to type and location are needed, as well as other studies to test anti-inflammatory and neuroprotective therapies.

## Figures and Tables

**Figure 1 biomedicines-11-00223-f001:**
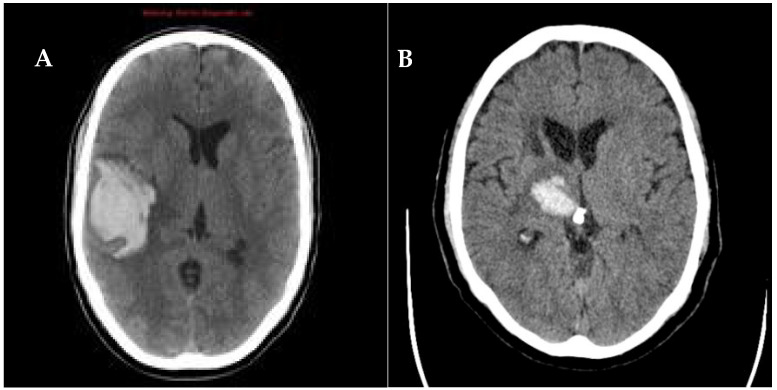
Lobar cerebral hemorrhage (**A**) and deep subcortical intracerebral hemorrhage of thalamic topography (**B**). Axial non-contrast brain CT scan.

**Figure 2 biomedicines-11-00223-f002:**
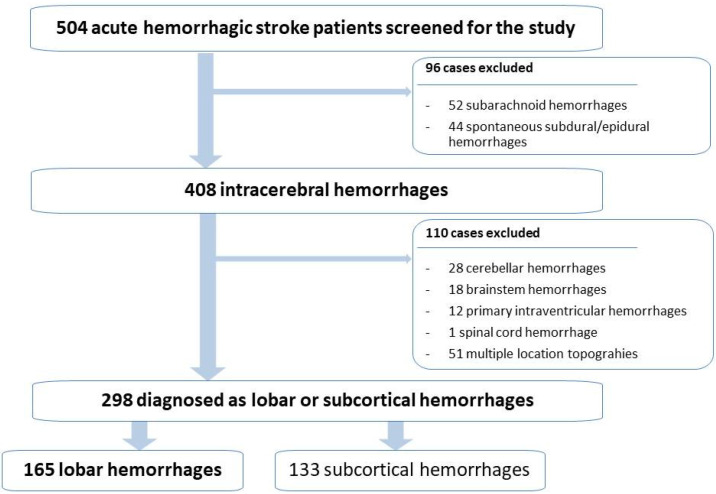
Schematic workflow of the study considering the different topographic subtypes of spontaneous acute intracerebral hemorrhagic stroke.

**Figure 3 biomedicines-11-00223-f003:**
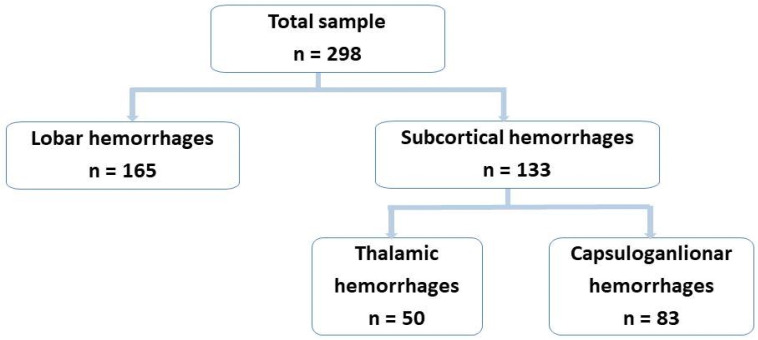
Topography of the spontaneous intracerebral hemorrhages included in the study.

**Figure 4 biomedicines-11-00223-f004:**
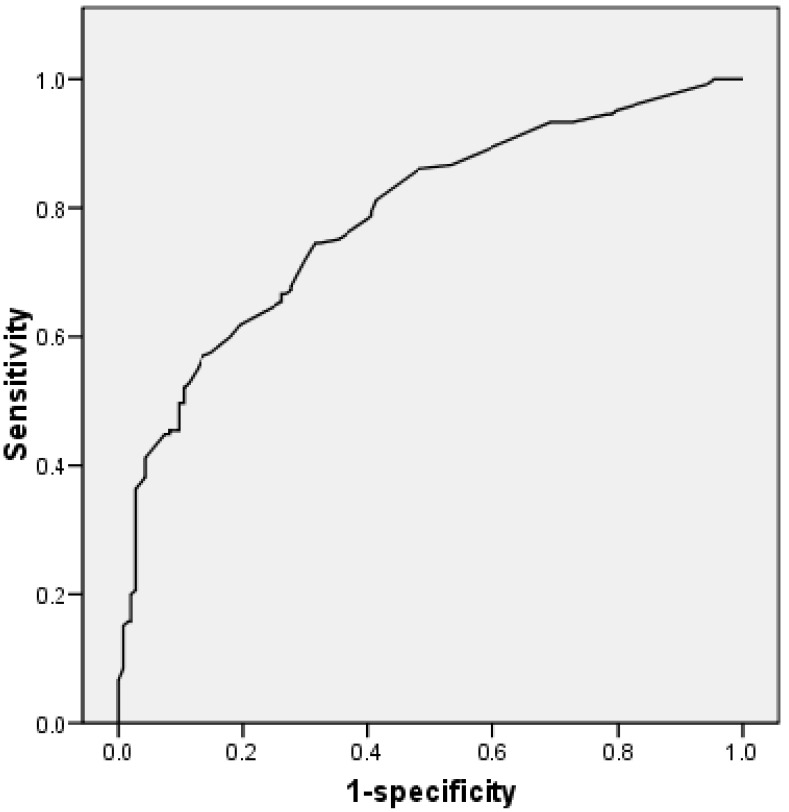
ROC curve for regression model including demographics, cerebrovascular risk factors, clinical features, and outcomes. Area under the curve = 0.785. ROC: receiving operating characteristics.

**Table 1 biomedicines-11-00223-t001:** Causes of acute spontaneous intracerebral hemorrhages.

Arterial Hypertension (Deep Perforating Vasculopathy)	Cerebral Amyloid Angiopathy
Acute arterial hypertension - Drugs - Cold exposition - Trigeminal nerve stimulation - Cardiac catheterization - Insect bites (wasp, viper) - Electroshock	Brain vascular malformations - Brain arteriovenous malformation - Cavernous malformation - Venous angioma - Dural arteriovenous fistula - Intracranial arterial aneurysm
Intracranial venous thrombosis (CVT) - CVT from local cause (head trauma, head and CNS infections, tumor) - CVT from systemic cause (pregnancy and puerperium, cancer, prothrombotic disorder)	Hemorrhagic transformation of cerebral infarction - Spontaneous hemorrhagic infarction - Hemorrhagic infarction occurring in patients receiving anticoagulant or thrombolytic therapy
Cerebral tumors - Primary tumor - Metastatic (mostly from melanoma, kidney and lung)	Hemostatic and hematologic disorders - Primary coagulopathy - Severe thrombocytopenia - Severe clotting factor deficiency such as hemophilia - Afibrinogenemia - Secondary coagulopathy - Anti-thrombotic drugs
Vasculitis and related vasculopathies - Systemic vasculitis - Isolated primary angiitis of the central nervous system - Reversible cerebral vasoconstriction syndrome - Posterior reversible encephalopathy syndrome - Infective endocarditis - Infections of CNS - Other vasculopathies	Changes in cerebral blood flow - Cardiologic surgery - Carotid endarterectomy or thrombectomy
Toxic - Cocaine - Other sympathomimetic drugs Rare entities - Dissection of intracranial arteries - Hyperperfusion syndrome	Other - Migraine - Alcohol abuse - Familial CAA - CADASIL COL4A1 mutations

**Table 2 biomedicines-11-00223-t002:** Demographic data of patients with spontaneous acute lobar cerebral hemorrhages versus deep subcortical intracerebral hemorrhages.

Variables	Lobar Intracerebral Hemorrhages (n = 165)	Deep Subcortical Intracerebral Hemorrhages (n = 133)	*p*
Female sex	89 (53.9)	51 (38.3)	0.007
Age (years), mean (SD)	72.48 (13.01)	72.23 (11.91)	0.867
Age ≥ 85 years	27 (16.4)	19 (14.3)	0.622

**Table 3 biomedicines-11-00223-t003:** Cerebrovascular risk factors, clinical features, and outcomes between patients with lobar versus deep subcortical intracerebral hemorrhages.

Variables	Lobar Intracerebral Hemorrhages (n = 165)	Subcortical Intracerebral Hemorrhages (n = 133)	*p*
Cerebrovascular risk factors (%)			
Hypertension	80 (48.5)	94 (70.7)	0.000
Diabetes mellitus	22 (13.3)	19 (14.3)	0.812
Valvular heart disease	7 (4.2)	3 (2.3)	0.344
Coronary heart disease	14 (8.5)	9 (6.8)	0.581
Atrial fibrillation	25 (15.2)	16 (12.0)	0.437
Congestive heart failure	3 (1.8)	4 (3.0)	0.500
History of transient ischemic attack (TIA)	11 (6.7)	7 (5.3)	0.613
Previous cerebral infarction	13 (7.9)	14 (10.5)	0.429
Previous intracerebral hemorrhage	19 (11.5)	5 (3.8)	0.014
Headache	8 (4.8)	2 (1.5)	0.111
Chronic obstructive pulmonary disease	13 (7.9)	13 (9.8)	0.564
Chronic renal disease	1 (0.6)	3 (2.3)	0.219
Peripheral vascular disease	8 (4.8)	9 (6.8)	0.478
Obesity	3 (1.8)	10 (7.5)	0.017
Alcohol abuse (>80 gr/day)	3 (1.8)	12 (9.0)	0.005
Anticoagulants	11 (6.7)	5 (3.8)	0.268
Chronic liver disease	16 (9.7)	5 (3.8)	0.046
Heavy smoking (>20 cigarettes/day)	16 (9.7)	11 (8.3)	0.670
Hyperlipidemia	19 (11.5)	18 (13.5)	0.599
Clinical features (%)			
Sudden onset	87 (52.7)	85 (63.9)	0.052
Acute onset (hours)	44 (26.7)	27 (20.3)	0.200
Subacute onset (days)	8 (4.8)	6 (4.5)	0.891
Headache	67 (40.6)	33 (24.8)	0.004
Dizziness/vertigo	7 (4.2)	2 (1.5)	0.170
Early seizures	15 (9.1)	2 (1.5)	0.005
Nausea/vomiting	34 (20.6)	23 (17.3)	0.470
Altered consciousness	71 (43.0)	39 (29.3)	0.015
Limb weakness	107 (64.8)	112 (84.2)	0
Sensory deficit	55 (33.3)	74 (55.6)	0
Hemianopia	52 (31.5)	20 (15.0)	0.001
Speech disturbances	55 (33.3)	49 (36.8)	0.528
Ataxia	9 (5.5)	5 (3.8)	0.492
Cranial nerve palsy	1 (0.6)	5 (3.8)	0.054
Extrapyramidal disorders	3 (1.8)	2 (1.5)	0.834
In-hospital outcomes			
Neurological complications	43 (26.1)	16 (12.0)	0.003
Respiratory complications	18 (10.9)	17 (12.8)	0.618
Digestive complications	5 (3.0)	3 (2.3)	0.681
Renal complications	1 (0.6)	6 (4.5)	0.027
Urinary complications	17 (10.3)	15 (11.3)	0.787
Cardiac events	5 (3.0)	6 (4.5)	0.500
Vascular complications	2 (1.2)	1 (0.8)	0.692
Hemorrhagic events	2 (1.2)	0 (0.0)	0.203
Infectious complications	25 (15.2)	34 (25.6)	0.025
Symptom free at discharge	10 (6.1)	11 (8.3)	0.459
In-hospital death	44 (26.7)	22 (16.5)	0.036
Length of stay, days, median (interquartile range)	15.00 (8–26)	18 (8–27)	0.938
Prolonged hospital stay > 12 days	96 (58.2)	86(58.2)	0.254

SD: standard deviation.

**Table 4 biomedicines-11-00223-t004:** Results of multivariate analysis independently associated with lobar hemorrhages in the different logistic regression models.

Variables	β	SE (β)	OR (CI 95%)	*p*
Demographic data and cerebrovascular risk factors ^1^				
Chronic liver disease	1.655	0.699	5.233 (1.328–20.610)	0.018
Previous intracerebral hemorrhage	1.117	0.538	3.056 (1.064–8.775)	0.038
Female sex	0.563	0.253	1.755 (1.070–2.879)	0.026
Hypertension	−0.916	0.259	0.400 (0.241–0.664)	0.000
Alcohol abuse (>80 gr/day)	−2.237	0.801	0.107 (0.022–0.513)	0.005
Demographic data, cerebrovascular risk factors, and clinical features ^2^				
Early seizures	1.776	0.858	5.906 (1.099–31.749)	0.038
Chronic liver disease	1.584	0.763	4.875 (1.093–21.745)	0.038
Previous intracerebral hemorrhage	1.176	0.595	3.240 (1.009–10.399)	0.048
Hemianopia	0.868	0.346	2.383 (1.210–4.693)	0.012
Headache	0.654	0.301	1.922 (1.066–3.467)	0.030
Altered consciousness	0.608	0.284	1.837 (1.053–3.206)	0.032
Limb weakness	−0.747	0.346	0.474 (0.241–0.933)	0.031
Sensory deficit	−0.814	0.285	0.443 (0.253–0.775)	0.004
Hypertension	−0.902	0.283	0.385 (0.221–0.670)	0.001
Alcohol abuse (>80 gr/day)	−2.171	0.849	0.114 (0.022–0.603)	0.011
Demographic data, cerebrovascular risk factors, clinical features, and outcomes ^3^				
Early seizures	1.918	0.855	6.806 (1.273–36.397)	0.025
Chronic liver disease	1.515	0.759	4.551 (1.027–20.154)	0.046
Hemianopia	0.937	0.358	2.551 (1.264–5.149)	0.009
Headache	0.644	0.298	1.904 (1.062–3.414)	0.031
Limb weakness	−0.744	0.342	0.475 (0.243–0.928)	0.029
Sensory deficit	−0.834	0.286	0.434 (0.248–0.761)	0.004
Hypertension	−0.902	0.281	0.406 (0.234–0.704)	0.001
Alcohol abuse (>80 gr/day)	−2.318	0.857	0.098 (0.018–0.528)	0.007

^1^ β = 0.432, SE(β) = 0.244, Hosmer–Lemeshow goodness-of-fit test χ^2^ = 0.958, df = 6, *p* = 0.987, correctly classified percentage = 63.1%. ^2^ β = 0.963, SE(β) = 0.370, Hosmer–Lemeshow goodness-of-fit test χ^2^ = 6.985, df = 8, *p* = 0.538, correctly classified percentage = 70.5%. ^3^ β = 1.065, SE(β) = 0.358, Hosmer–Lemeshow goodness-of-fit test χ^2^ = 5.384, df = 8, *p* = 0.716, correctly classified percentage = 69.5%.

## Data Availability

Not applicable.

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
