# Peer review of "Acute Spontaneous Lobar Cerebral Hemorrhages Present a Different Clinical Profile and a More Severe Early Prognosis than Deep Subcortical Intracerebral Hemorrhages—A Hospital-Based Stroke Registry Study"

_biomedicines, 2023, doi:10.3390/biomedicines11010223_

Round 1
Reviewer 1 Report
Interesting and well written study regarding the pattern of cerebral hemorrhage and its correlation with clinical outcome.
the study is well conceived and presented.
discussion section is placed before metheds and results, this must be corrected.
discussion section as well as introduction are a bit too long.
I would suggest to add a comment regarding the role of different types of hemorrhages in AVM (look for e amplie: Sturiale CL, et al. Relevance of bleeding pattern on clinical appearance and outcome in patients with hemorrhagic brain arteriovenous malformations. J Neurol Sci. 2013 Jan 15;324(1-2):118-23. doi: 10.1016/j.jns.2012.10.016. Epub 2012 Nov 9. PMID: 23146614.)
Finally better outline the conclusions of this study at the end of the work.
Author Response
Dear reviewer:
1.We appreciate your supportive comment regarding the pattern of cerebral hemorrhage and its correlation with clinical outcome.of the study.
2.We have placed the Discussion section after Methods and Results.
3. We have add in the Discussion a comment regarding the role of different types of hemorrhages in AVM:
“In a clinical study Sturiale et al emphasized the possibility that in patients with hemorràgic brain arteriovenous malformation non-parenchymal bleeding (subarachnoid hemorrhage and intraventricular hemorrhage) may worsen the outcome of patients with hemorrhagic arteriovenous malformations”.
4. The reference suggested by the reviewer is also included (reference 71): Sturiale CL, Puca A, Calandrelli R, D'Arrigo S, Albanese A, Marchese E, Alexandre A, Colosimo C, Maira G. Relevance of bleeding pattern on clinical appearance and outcome in patients with hemorrhagic brain arteriovenous malformations. J Neurol Sci. 2013 Jan 15;324(1-2):118-23. doi: 10.1016/j.jns.2012.10.016. Epub 2012 Nov 9. PMID: 23146614.
5. We have better outline the conclusions of the study and at the end of the work we have adding this sentence: “ICH is a complex clinical event that has been shown to benefit from specially trained multidisciplinary care. Trials of minimally invasive surgical evacuation of clots according to the type and location are needed, as are other studies to test anti-inflammatory and neuroprotective therapies”.
We thank the reviewer for his time and helpful suggestions to improve the quality of the manuscript.
Reviewer 2 Report
A paper is about a very polymorphic group of patients with a spontaneous supratentorial intracerebral hemorrhage. The structure of the paper is strange - results are presented above the description of the study population and methods. Consider changing the structure.
Patients with a secondary bleeding due to vascular malformations were included into the study, however treatment and outcomes in this group of patients are very different: bleeding from e.g. ruptured cavernous malformation is less life-threatening (brainstem is excluded) and treatment is usually surgery with a better outcomes. Consider adding the description of study group and treatment.
It is not clear, which demographic, cerebrovascular risk factors, clinical features and outcome data was used in figure 2. Is this sensitivity and specificity for outcome (bad outcome of good outcome)? Consider clarification of the data and calculations.
Author Response
Dear reviewer:
Thank you for your valuable comments and suggestions.
1.We have considered changing the structure: we have placed the Discussion section after Methods and Results.
2.According to the valuable suggestion of the reviewer we have expanded the text by adding another limitation of our study: “Another limitation of the study is that secondary bleeding due to vascular malformations (7.5% of the sample) were included into the early outcome. However treatment and early prognosis in this group of patients are very different because bleeding ruptured cavernous malformation is less life-threatening (and treatment is usually surgery with a better outcomes) than macrovascular lesions such as arteriovenous malformations and aneurysms [30]”.
3.In accordance with the reviewer’s suggestion, in the revised version of the manuscript, we have clarified that this figure shows the ROC curve for the accuracy of the regression model including demographic data, cerebrovascular risk factors, clinical features, and outcome on multivariate analysis. The area under the curve was 0.785. The sensitivity was 66.7%, the especificity was 72.9%, the positive predictive value was 75.3%, and the negative predictive value was 63.8%. In the multivariate analysis, the cerebral topography (coded as subcortical location=0 and lobar location =1) was the dependent variable (please see Matherials and Methods section).
We wish to thank the reviewer for his valuable and supporting comments.
Round 2
Reviewer 2 Report
Corrections are adequate